# Open Data Policies among Library and Information Science Journals

**Brian Jackson** [ID]

Library, Mount Royal University, 4825 Mount Royal Gate SW, Calgary, AB T3E 6K6, Canada; bjackson@mtroyal.ca

**Abstract:** Journal publishers play an important role in the open research data ecosystem. Through open data policies that include public data archiving mandates and data availability statements, journal publishers help promote transparency in research and wider access to a growing scholarly record. The library and information science (LIS) discipline has a unique relationship with both open data initiatives and academic publishing and may be well-positioned to adopt rigorous open data policies. This study examines the information provided on public-facing websites of LIS journals in order to describe the extent, and nature, of open data guidance provided to prospective authors. Open access journals in the discipline have disproportionately adopted detailed, strict open data policies. Commercial publishers, which account for the largest share of publishing in the discipline, have largely adopted weaker policies. Rigorous policies, adopted by a minority of journals, describe the rationale, application, and expectations for open research data, while most journals that provide guidance on the matter use hesitant and vague language. Recommendations are provided for strengthening journal open data policies.

**Keywords:** open data policies; research data management; academic publishing; library and information science

## 1. Introduction

The idea of research data as a public good is burgeoning across disciplines. In this shifting conception, open research data is observed as bolstering the transparency and reproducibility of results reported in other research products and as a distinct contribution to the scholarly record. While datasets may be standalone outputs of a research endeavor, more often they are associated with other research materials, typically journal articles. Regardless of whether the intended use of open research data is for secondary analysis or monitoring the accuracy of reported findings, there is no doubt that publishers of academic journals have a significant role to play in supporting open research data initiatives [1]. A key part of that role is the development of open data policies to promote the public archiving of data that support the results of published research.

Open data policies lay out the expectations of a journal for the management and sharing of research data underpinning findings reported in the publication. Those policies may also describe how authors should communicate the status and availability of research datasets to readers, usually through data availability statements. Open data policies vary greatly in terms of rigidity, precision, and scope, and have by no means been universally adopted. Despite these variations, the implementation of journal data policies is touted as a useful strategy for promoting open data at a key point of intervention in the research process.

In academic institutions, librarians have also played a role in promoting open data through the development of various research data management services, including the establishment of repositories that support open data archiving [2]. Graduate-level library and information science (LIS) curricula increasingly include topics related to data curation and management [3]. Because of the degree to which the topic has seeped into the work and

training that occurs in the field, many practitioner and academic researchers working in the LIS discipline have been exposed to information about research data management and the principles of open research data, some of whom have developed significant expertise.

Many LIS researchers are also engaged with academic publishing, broadly and within the discipline, as consumers, contributors, and providers of editorial oversight [4]. That the discipline has feet firmly planted in the arenas of open research data and academic publishing suggests that LIS journals might be keen adopters of open data policies. However, journals in the field are managed by publishers and associations with varying distribution models, access to resources, content types, and subject foci. Because of this variation, it is not a certainty that open data policies are overly prevalent among LIS journals. This study will examine the adoption of data policies among journals in the field.

A growing body of research has examined the relationships between journal publishing and open data practices. One category of research in this area looks at the frequency with which researchers publicly archive datasets that support the findings reported in journal literature. Research has examined data availability rates in journals from a variety of disciplines, including biology, chemistry, mathematics, and physics [5], biomedical disciplines [6], neurology [7], ecology and evolution [8], political science [9], and addiction studies [10], among others. The rates at which data is made publicly available in these disciplines ranges from less than 1% to 58%, with most falling between 10% and 20%.

A related thread of inquiry, like the present study, looks at the nature of, and extent to which, journals have adopted data archiving or publishing policies across disciplines. Among recent studies, Vasilevsky et al. found that only a minority (11.9%) of more than 300 biomedical journals required data sharing as an explicit condition of publication [11]. While a larger number of journals in the study encouraged sharing, nearly 32% of journals did not mention the practice. Choi et al. found that 29% of 781 academic society journals, across disciplines, adhered to the Committee on Publication Ethics' guidelines for data sharing and reproducibility [12]. Among science journals published in South Korea, 13% of surveyed editors claimed to have adopted a data sharing policy [13]. Rousi and Laakso found higher rates of open data archiving policies among neuroscience (87.5%), physics (65%), and operations research (77.5%) journals, although the reported figures include all types of policies, including those that encourage, but do not require, data sharing [14]. Fewer journals in that study required data availability statements as part of open data policies (55%, 37.5%, 60%, respectively). Johnson et al. found that only 48% of neurosurgery and neuroscience journals had adopted some form of data sharing policy, with 21% encouraging data availability statements [7].

In addition to differences in disciplinary practices, variability within the details of data sharing policies, and their interpretation, may account for the wide range of reported rates of policy adoption. The Jisc Journal Research Data Policy Registry pilot was an attempt to develop a central repository of open data policies, but it was ultimately found unfeasible due to a lack of standardization and harmonization across policies [15]. In one qualitative study, Sturges et al. found inconsistencies in open data policies within and between journals, including ambiguity among definitions of the types of data that require open deposit, where data should be archived, the level of third-party access required by policies, and timelines for deposit [16]. Some journal websites contained more than one open data policy. Christiansen et al. compared the text of data policies with the interpretations of and experiences with those policies among authors and editors, finding inconsistencies even between editors' descriptions and the contents of policies [17]. Clarity of data policies appears to be influenced by disciplinary norms, as journals publishing in fields with established data practices tend to have more detailed policies [14].

There have been attempts within some organizations to develop a consistent approach to open data policies. Some large publishers have standardized open data policies for journals under their banners. Both Taylor & Francis and Springer Nature, for example, have adopted tiered data policy descriptions, ranging from encouragement to strict requirement of open data deposition that may be adopted by their journals [18,19]. Academic

publishing associations, and other related bodies, have also issued guidelines that outline preferred practices for data citation and sharing. Examples of these include the Committee on Publication Ethics' (COPE) *Principles of Transparency and Best Practice in Scholarly Publishing*, the *Guidelines for Transparency and Openness Promotion (TOP) in Journal Policies and Practices*, a joint statement on data accessibility by the International Association of Scientific, Technical and Medical Publishers (STM), the Association of Learned and Professional Society Publishers (ALPSP), and the *Force 11 Joint Declaration Of Data Citation Principles*. While many journal publishers have expressed support for and adopted elements of these guidelines, their direct impact on data transparency is difficult to measure.

There is some evidence that open data policies do have an impact on data archiving practices, although there are mixed results when examining overall effectiveness in improving quality and access to data. Some studies have found, unsurprisingly, that open data deposition does increase among authors publishing in journals with robust open data policies [9,20–22]. These effects may be strengthened with the promotion of other incentives for publishing data, such as increased citations [17,23], badges [24], and awards [25]. Doubts have been raised, though, that compliance with existing policies and quality of archived data are optimal. Federer et al. found that only 20% of data availability statements in *PLoS One* articles pointed readers to datasets available in an open repository, the journal's preferred method of data sharing [26]. Other research that has examined data availability found incomplete and unusable data and challenges in contacting authors who previously indicated that data would be made available upon request [8,27–29]. As a potential solution, much of this research points to the (re)development of data sharing policies to ensure that they are consistent, detailed and specific, unambiguous, and stringent, while highlighting incentives for open data publishing [8,11,17,26].

In the library and information science discipline, Aleixandre-Benavent et al. examined the relationship between journals' impact factors and the presence of open policies, finding a positive correlation [30]. This result is consistent with findings from other fields [16,31]. To date, though, there has not been a close examination of open data policies in LIS journal publishing. As open data practices are often tied closely to disciplinary culture, and with the library and information community's significant involvement in advocacy efforts to advance open data, it is important to take stock of the prevalence and nature of open data policies in LIS literature in order to identify paths forward for the discipline. Further, as data sharing behaviors among researchers are associated with open access publishing experiences [32], examining connections between publishing models and open data policies may provide insights into patterns that can lead to increased data availability.

For that reason, this study will examine information provided on the websites of library and information science journals in order to measure how widely open data policies have been adopted by these journals, particularly in relation to journal publishing models, and examine the nature of the guidance provided on data management to prospective authors.

## 2. Materials and Methods

A LIS journal title list was sourced in June 2020 using the Journal Citation Reports (JCR) and Scopus databases. The initial list contained 89 titles from JCR's Information Science and Library Science category and 204 titles from Scopus's Library and Information Science subject group. Only journals actively publishing original research articles, and that maintain a website primarily in English at the time of data collection, were included in the study. After these criteria were applied and duplicate journals removed, the final list contained 201 journal titles.

Due to the nature of categorization employed by JCR and Scopus, the journals included in the study represent a variety of sub-fields within the larger LIS discipline, including public, academic, and special library science, archival studies, publishing, information systems, records management, and disciplinary-focused information studies, in addition to a large number of generalist LIS journals. The geographical representation of the sample

is narrower, with 73% of the journal list being published in either the United Kingdom (76) or the United States (71). The remaining journals are published in the Netherlands (18), Canada (6), Germany (5), Australia (3), and other countries throughout Europe, Asia, and Africa.

The website of each journal was reviewed for policies or guidance documentation pertaining to open research data, beyond those that addressed only supplementary data. The text of research data policies, editorial expectations for the management and provision of research data, data availability statement templates and descriptions, and other information that described the journal's standards for the management of research data were copied from the journal website into a static document for analysis. Publisher-provided guidance on the handling of research data was only recorded if copied on the journal website. Although contributors may encounter publisher policies not available on journal websites during the publishing process, the goal of this exercise was to examine how the journals represent their expectations for research data on their own publicly facing websites. Finally, each journal was classified as open access, defined as providing free access to all current and archival issues, or not.

A mixed methods approach was used to analyze the data. Each journal's policy related to research data was coded within two categories: data availability—the degree to which publishers require authors to provide public access to data—and data statements—the presence of instructions that authors describe if, and where, data is available. Within the data availability category, each journal received a code of required, suggested, conditional (for instances in which data must be made available when conditions require it, as when it is requested), or no information provided. Within the data statement category, each was coded as required, suggested, or no information was provided.

A thematic analysis was also applied to the information recorded from each journal's website, using an inductive approach based on Braun and Clarke [33]. All text copied from each website was grouped by journal, without separation based on location, within the website. An initial reading of the complete body of text was conducted in order to develop familiarity with the data. During the second and third readings, codes were constructed using a semantic approach, examining the topics related to research data contained in each text. Coding during the third reading was conducted without reference to that of the second reading. Codes developed during both readings were then compared against each other and against the text, followed by the merging of like codes and thematic grouping. While this process offers some degree of intracoder reliability, it should be noted that the final constructions are heavily informed by the researcher's interpretations and experiences, which may be considered a limitation of the study. While the themes were developed from within the texts provided by journals, analysis of those themes was conducted within a broader context, including documents referenced by the research data approaches of the journal.

## 3. Results

### 3.1. Quantitative Analysis

Of 201 journals, 10 (5%) explicitly required original research data to be made publicly accessible in an open archive, while 6 (3%) required that data be made available conditionally upon request. A further 96 (48%) recommended that datasets be publicly archived but did not mandate open archiving as a condition of publication. Information on data availability requirements was absent from the websites of 89 journals (44%).

Data availability statements were a requirement of 16 (8%) journals and a recommendation of 58 (29%) others. Most journals in the study (127, 63%) did not indicate a requirement or suggest that such statements be included with article submissions.

Of the 201 journals, 32 were classified as fully open access. Although the number of open access journals was small relative to the total sample, they accounted for 6 of the 10 journals that require research data to be openly archived. A further 3 open access journals require conditional data access, while 3 encourage open data archiving.

Commercial journals were far more likely to recommend that datasets be openly archived than they were to require the practice. Of 169 subscription journals, only 4 required open data availability, while 93 encouraged authors to openly archive data. An additional 3 journals required conditional data access.

No country of publication was significantly over- or underrepresented among journals that require open data practices. Of the ten, eight are published in the United Kingdom (3), the Netherlands (3), and the United States (2), the three most represented countries in the overall sample, with two others published in Switzerland and Iran. Journals requiring open data archiving do, however, represent a relatively narrow range of sub-fields within the LIS discipline, including discipline-focused information studies (chemistry, geography, linguistics), publishing, archival science, and medical librarianship.

*3.2. Qualitative Analysis*

Thematic analysis of the textual information resulted in the identification of 12 codes. Those twelve codes were categorized into four major themes that describe the nature of information that LIS journals provide about research data policies—data management, procedural guidance, principles, and policy limitations. Table 1 outlines the thematic groupings.

**Table 1.** Major themes and associated codes developed through qualitative analysis of library and information science (LIS) journal research data guidance.

| Major Theme | Codes |
| --- | --- |
| Data Management | File formats, licensing, persistent identifiers, data retention |
| Procedural Guidance | Defining data, templates, repositories |
| Principles | Justification, organizational linkage, data citation |
| Policy Limitations | Privacy, limiting language |

3.2.1. Data Management

The theme of research data management practices occurs frequently in journal guidance documentation. This theme most often occurs as recommendations or specifications about the packaging and handling of research data that support published papers. Among the topics within this theme, the use of persistent identifiers for archived data is the dominant area of concern for LIS journals. In nearly all cases in which journals mandate or recommend archiving research data, linking to datasets using persistent identifiers is a requirement. Digital object identifiers, and persistent URLs more generally, are the most frequently specified methods of linking to datasets. Most journals leave the reasoning for this requirement unstated, but some do provide a justification. For example, nearly every Emerald Publishing Group journal explains that "persistent identifiers ensure future access to unique published digital objects, such as a piece of text or datasets", while referring to both the TOP Guidelines and the Data Preservation Alliance for the Social Sciences (Data-PASS) group.

Only a small minority of journals provide information about formatting datasets. In some instances, the context of these recommendations suggests that they are there to align with and promote good data management practices, but for some journals, file formatting guidance likely stems from the preferred practices of repositories linked to the journal. Article submission procedures for *Evidence-Based Library and Information Practice*, for example, are synchronized with data deposits to the University of Alberta Dataverse repository. That journal recommends that data files be in "non-proprietary, unencrypted, uncompressed formats" and provides references to best practices documents for data file formats.

Although many journals in this study require or recommend that research data be made openly available, only one was specific about the type of license that should be applied to third-party use of the data. The *International Journal of Geographical Information Science*, published by Taylor & Francis, is the only journal among 45 Taylor & Francis

journals in this sample to use the publisher's Open and FAIR data sharing policy. That policy requires archived datasets to be archived with either a Creative Commons No Rights Reserved (CC0) or Creative Commons Attribution (CC-BY) license. Although other journals distributed by Taylor & Francis and other publishers use terms such as "open" or "freely available" to describe preferred access levels, none were specific about the terms under which data should be licensed.

A small number of journals, all of which require data to be made available only upon request, discuss data retention in their guidelines. In each case, an appropriate data retention period is left unspecified—authors are instructed to retain data for a "reasonable" time period so that they can be made available if requested.

### 3.2.2. Guidance

The volume of pre-submission procedural guidance for data archiving varies significantly by journal and publisher. A foundational element that is absent from a majority of journal websites is a definition of research data. Elsevier is the only major publisher that frames the concept of data, but the publisher's definition is broad and open to interpretation: "Research data refers to the results of observations or experimentation that validate research findings". A small number of association and independent publishers provide more specific examples. The *Journal of the Medical Library Association*, for example, applies its policy to "authors of Original Investigation, Case Report, and Special Paper articles" and defines data as "the digital materials underlying the results described in the manuscript, including but not limited to spreadsheets, text files, interview recordings or transcripts, images, videos, output from statistical software, and computer code or scripts". That journal also advises authors to deposit at least the minimum amount of data required to reproduce the results described in related articles.

A greater number of journal websites recommend data repositories, although relatively few of these provide details about those repositories and the terms under which they operate. The generalist repositories Dryad and figshare are recommended most commonly, but some websites contain or link to longer lists of potential repositories. In many cases, for both larger publishing houses and independent journals, specific repositories are recommended due to existing collaborations. Journals published by Elsevier highlight the Elsevier-operated Mendeley Data repository, but do not require authors to use that service exclusively. Sage journals similarly promote the publisher's relationship with figshare. For independent journals and smaller publishers, repository connections primarily stem from journal-specific spaces in larger repositories. *LIBER Quarterly*, for example, recommends its dataverse in the Harvard Dataverse repository, while *Evidence-Based Library and Information Practice* integrates data deposit in the University of Alberta Dataverse into the submission process.

Guidance on the formatting of data availability statements is common among LIS journals. There is a distinct difference between journals that require open data archiving and those that recommend the practice. Journals that require data deposit more commonly outline specific requirements for data availability statements, which usually include the name of the repository, where the data are located, and a persistent identifier. Some journals in this subset have more specific guidelines, providing templates or a list of statements from which authors may choose the one that best describes the status of their data. Journals that only recommend data deposit, on the other hand, are more likely to require data availability statements only when authors make the decision to archive data. For example, Taylor & Francis journals that apply the publisher's basic data sharing policy, the largest group of journals in this sample, instruct authors that "if there is a data set associated with the paper, please provide information about where the data supporting the results or analyses presented in the paper can be found", while providing a selection of statement templates. A small number of journals, not requiring deposit, do ask that authors provide some statement of explanation if authors choose not to do so. *Insights: The UKSG Journal*, for example, provides the following guidelines, "If data is not being made available with

the journal publication, then ideally a statement from the author should be provided within the submission to explain why".

### 3.2.3. Principles

Only a small number of journals provide a rationale for their open data policies and practices. In most of those cases, reasons for the journal's support for open data are very brief, often taking the form of succinct preambles to procedural or guiding documentation. The journal *Publications*, for example, begins its section on open data with the statement, "In order to maintain the integrity, transparency and reproducibility of research records, authors must make their experimental and research data openly available". In some cases, journals provide a statement of principles, anchored to open data practices by proximity rather than through explicit cause and effect statements. Open data guidelines provided by Sage journals open with a standalone statement of the publisher's open data ethic not directly connected to the guidelines: "At SAGE we are committed to facilitating openness, transparency and reproducibility of research. Where relevant, The Journal encourages authors to share their research data in a suitable public repository". *Evidence-Based Library and Information Practice*, by contrast, clearly links its open data policy to a justification for that policy, outlining numerous goals and benefits of open data.

Other journals affirm commitments to open data by expressing support for the statements of external stakeholders or by noting adherence to the open data practices of their publishers. This type of support via organizational linkage is most common among the journals of major publishers but occurs in statements from independent and small-publisher journals, as well. While the open data guidelines provided by Emerald journals are relatively scant, each journal emphasizes support for the *Transparency and Openness Promotion Guidelines*, an inter-organizational set of principles for open research coordinated by the Center for Open Science. Some journals copy statements related to data access directly from, or express adherence to, guidelines provided by publishing industry organizations, including the International Organization for Science, Technical, and Medical Publishers (STM), the Association of Learned and Professional Society Publishers (ALPSP), and the Committee on Publication Ethics (COPE).

In addition to these external guidelines and documents, a small number of journals refer to the *Force 11 Joint Declaration of Data Citation Principles* and the Data-PASS citation guidelines when describing editorial policies on data citation. While specific reference to these documents occurs relatively infrequently within the sample, instructions on citing both author-developed datasets and secondary datasets are very common. Among the journals that do not explicitly reference either the Force 11 or Data-PASS documentation, the data citation practices prescribed on their websites largely adhere to the practices recommended by those groups.

### 3.2.4. Policy Limitations

Nearly all journals in the sample hedge requirements or recommendations for open data archiving using limiting language. For some journals, particularly those that only encourage open data archiving, specific reasons for exemption are left unstated. As examples, Elsevier journals explain that "This journal encourages and enables you to share data that supports your research publication *where appropriate* [italics added]", while some Springer journals state, "The journal encourages authors, *where possible and applicable* [italics added], to deposit data that support the findings of their research in a public repository" without further explanation. Most journals that have adopted mandatory open data requirements also outline limitations to the policy. One notable difference between journals that mandate open data and those that recommend the practice is the placement of limiting language in the policy text. While many journals that encourage data archiving begin policy descriptions using hedging language as the above examples illustrate, journals that require open data are more likely to begin with firm statements and acknowledge exemptive criteria further down the text of the policy.

When journals do address specific reasons that data sharing requirements may be waived, they most frequently discuss the impracticality of sharing some types of data, ethics and privacy concerns, and proprietary data. Few provide details about the nature of these limitations, although a small number do link to additional policies on publication ethics or explicitly state that data should be anonymous or de-identified. Limiting language also extends to the conditions under which authors are expected to include data availability statements. When data will not be publicly archived, journals with strict data policies require that any restrictions on data access be justified in the statement, but most simply ask authors to state that data are unavailable. Some of those with more detailed policies provide templates for such scenarios, such as a small number of Springer journals that differentiate between categories of data access limitations in data availability statement guidelines.

## 4. Discussion

Findings from both the qualitative and qualitative analyses highlight differences in the approaches to open data policies among large commercial publishers versus that of independent or open access publishers. Of 201 journals in this study, 116 are published under the banner of five large publishing groups: Taylor & Francis/Routledge, Elsevier, Emerald, Sage, and Springer. In terms of articulating and encouraging open data practices by authors in the discipline, the effect of a dominant presence of large commercial publishers is mixed.

Journals published by these companies are far more likely than independent or small publisher journals to address open data on their websites. Most follow a template in which they expressly encourage authors to archive and adequately cite data, provide information about repository options, briefly describe the principles underpinning the open research data movement, and provide guidance on crafting data availability statements. Because of their ability to reuse guiding documents across multiple journals, large publishers have made this approach to framing open data archiving common among LIS journals. Some parent publishers also provide additional, detailed information about open data best practices through their own websites, but direct linking to these resources by individual journals is inconsistent.

The incorporation of existing guidelines into the open data policies of commercial journals also represents a significant strength. These policies lean heavily on various statements of principles, and best practices, distributed by publisher associations and other groups invested in promoting open data. Reliance on established practices encourages consistency of messaging and approaches to data management, while providing a frame of reference through which editorial staff and authors may establish a shared understanding of expectations.

While the size and resource advantages of large publishers have allowed them to amplify generalized support for open data, there may be drawbacks to this outsized influence within the discipline. Most of the support for open research data expressed by commercial journals ends at encouragement and basic guidance. Even where tiered open data policies have been developed at the publisher level, the vast majority of journals in the discipline have adopted weaker policy options. This predilection for lower-tiered policies may have a significant impact on the overall availability of data stemming from research in the discipline. According to Vines et al., data stemming from research published in journals that employ strict open data mandates are 17 times more likely to be publicly available when compared to research data from journals without open data policies, while data availability rates among journals that merely encourage data sharing are only 3.6 times greater [22]. Further, data availability for articles published in journals with both mandatory data archiving and data availability statement policies were 974 times greater than those without a policy.

According to Hrynaszkiewicz et al., Springer-Nature journals "adopt the data sharing policy that is most appropriate to its research community and the resources available to that community—encouraging the most relevant good practice for their community" [34](p. 68).

It is not clear why LIS journals have predominantly adopted weaker data sharing policies when given a choice, but perceptions about the discipline's research cultures, research methodologies, and established publishing practices may have a role. In a 2014 study, for example, only 16% of the articles published in a sample of LIS journals were classified by the authors as research articles [35]. Still, with data management services continuing to grow in academic libraries [36], researchers in the field may be more socialized to the idea of open data archiving than those working in other disciplines, and may be better prepared to comply with stricter policies.

Although most open access journals do not mandate open data archiving as a condition of publication, or address the topic at all on their websites, they account for more than half of LIS journals that do require the practice. Given the overlapping ethos underlying open access publishing and data sharing, this is not surprising. While some open access journals are affiliated with large publishers, most are published by smaller publishers or societies who may lack the resources to fully develop, and enforce, editorial policies in emerging areas. Should open data mandates become more common among LIS journals, there may be additional incentives and fewer barriers among small and independent journals to adapt developed and tested data sharing policies.

Absent from most journal websites in this study's sample are finer details about the principles behind open data policies, how and to which data these policies apply, and data management and formatting practices that lead to high quality open data. While it may be argued that the principles of open data policies are self-evident, and good research data management practices may be learned elsewhere, the inclusion of both of these types of information alongside policy details would serve to clarify the intent of the policy and expectations for data quality. Perhaps most importantly, strong policies should define as precisely as possible how they are applied. Given the breadth of data types involved, and the privacy and intellectual property considerations that may factor into open data practices, it is understandable that journals may be reluctant to narrowly address open data policy application. However, less ambiguous, strictly defined open data policies do improve data sharing rates, while reducing the burden of uncertainty for prospective authors. Based on these findings, it is recommended that LIS journals adopt more rigorous open data policies that incorporate the policy elements in Table 2 to the greatest possible degree.

**Table 2.** Recommended journal open data policy elements.

| Category | Recommended Policy Elements |
|---|---|
| Application | • Describe precisely the types of data to which open data policies apply;<br>• Target policies at a clearly defined body of research (e.g., quantitative data, original research, or methods-based requirements). |
| Exemptions | • Identify the circumstances that warrant exemptions as narrowly as possible;<br>• Describe valid concerns related to privacy or proprietary data;<br>• Require researchers to justify exemptions during the submission process. |
| Timing | • Indicate whether data should be publicly accessible prior to submission, prior to publication, or within a set time period after publication;<br>• Indicate if datasets will be considered in the peer-review process. |

**Table 2.** *Cont.*

| Category | Recommended Policy Elements |
|---|---|
| Licensing and Access | • Prescribe specific open licensing terms that authors should apply to datasets;<br>• Describe how authors should proceed if repository policies do not accommodate the preferences of the journal;<br>• If access to data will be justifiably restricted, require authors to detail any conditions under which they will be made available. |
| Formatting and Repositories | • Encourage authors to use preservation-friendly formats for data, with examples provided based on data type;<br>• Identify preferred data repositories;<br>• If journals do not specify data repositories, identify required characteristics of eligible repositories, including preservation and persistent linking features. |
| Data Availability Statements | • Require data availability statements for all eligible articles;<br>• Identify each element required in statements, such as location of and links to data, full citations, and terms of access;<br>• Require a specific justification when data are not available, rather than providing templates. |
| Principles | • State the underlying principles of the open data policy;<br>• Clearly outline the potential contributions of open data to the transparency and reproducibility of research, the potential for secondary analysis, and the ethos of research data as a public good. |

The adoption of stricter data archiving policies, particularly those that include mandated data availability statements, has been demonstrated to lead to much greater rates of public availability of research data. Open data policies have the added benefit of increasing citations for associated articles [37], benefitting both journals and authors. While this study has demonstrated a greater rate of adoption of open data policies among open access journals, there is room for growth in this area among both open access and commercial journals. Open access journals, most of which are published by smaller organizations, may consider examining their capacity to develop and support open data policies, even in a limited fashion. Even policies that only recommend open data archiving do impact data availability rates, if at much lower levels. Models for these policies are readily available [16], and authors who seek out open access journals may be more likely to comply with open data policies [32]. Journals published by commercial publishers may be limited by parent publisher policies, but those with tiered data policies options may look at adopting higher-tiered policies. Enhanced data sharing policies, at the publisher level, are likely to lead to a greater impact of journals under those banners and grow the data repositories with which some are affiliated.

While this study has examined the nature of open data policies among journals in the LIS discipline, it has done so in a broad way. The discipline encompasses a large number of sub-fields within it, each of which may have differing research practices. A closer examination of the ways in which research data are managed among the many facets of the LIS discipline, including subject focus and scope of journals, may lead to greater insights about how open data may be encouraged and promoted. Absent from this study is information about compliance with open data policies, or voluntary open data archiving in the absence of policy, within the discipline. Further study of the practices of LIS researchers may help journals adopt open data policies that align with the goals of the authors, readers, and the journals themselves.

## 5. Conclusions

While more than half of the library and information science journals in this study include information on their websites about expectations for open research data, it is primarily large commercial publishers doing so. Most journals that have adopted data policies encourage, but do not require, authors to openly archive data, using tentative and imprecise language on journal websites to describe the application of the policy. Across disciplines, compliance with open data policies is sub-optimal, with many authors using exemption options or simply failing to comply with sharing policies. Highly visible and specific research data policies, posted to journal websites, may help clarify expectations around data sharing for authors early in the journal selection process and enable more adequate planning for data archiving. Greater precision in the application of data policies may also free journals to make the practice a requirement in well-defined circumstances, while still allowing reasonable flexibility when the ability of researchers to share their data is limited.

**Funding:** This research received no external funding.

**Institutional Review Board Statement:** Not applicable.

**Informed Consent Statement:** Not applicable.

**Data Availability Statement:** The data presented in this study are openly available in the Mount Royal University Data Repository at https://doi.org/10.5683/SP2/TQLZ6G.

**Conflicts of Interest:** The author declares no conflict of interest.

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
