# Peer review of "Open Data Policies among Library and Information Science Journals"

_publications, doi:10.3390/publications9020025_

Round 1

Reviewer 1 Report

This is a well written paper discussing an important (and emerging) topic on open data policy in journals. The paper is logically written using well designed methods to answer the research problems. I only have minor suggestions. 1) Would it be possible to share the list of journals with open data policy as an appendix? I know it's a long list but it would be really helpful for other researchers to know what kinds of LIS journals the author is talking about. 2) It would be also helpful to know more information about the journals. The author clearly described the selection process for journals for this research but did not provide information about which journals are from US, Canada, or UK (given they are all English based journals), and the scope of journals (domestic, international). Even they are all English based journals, there may still be an effect of domestic context, so it would be interesting to know how many journals are US based, etc. 3) How did the author define "LIS"? As appeared and/or classified in SCOPUS? This seems to be important to claim any finding in the context of LIS, and even within LIS, depending the nature and the focus of journals, each may have a different attitude toward open data policy. While what the author has presented in the results are meaningful, I think adding more contextual information about the journals would be another unique contribution to the field.  4) Lastly, it would be nice if the author can extend the discussion on the implications. Using all the findings, where should we go? What would the author suggest to do to make the existing situation better? Practical recommendations would be very helpful.

Author Response

Response to Reviewer 1 Comments

I would like to thank the reviewer for their time and insightful review. My responses are below:

1) Would it be possible to share the list of journals with open data policy as an appendix? I know it's a long list but it would be really helpful for other researchers to know what kinds of LIS journals the author is talking about.

I have placed the full list of journals in an open repository, but would be happy to add it as an appendix or as supplementary data.

2) It would be also helpful to know more information about the journals. The author clearly described the selection process for journals for this research but did not provide information about which journals are from US, Canada, or UK (given they are all English based journals), and the scope of journals (domestic, international). Even they are all English based journals, there may still be an effect of domestic context, so it would be interesting to know how many journals are US based, etc. 

This is a potentially interesting factor. I have added a summary of the publishing locations of the journals. Nearly three-quarters are from the UK and the US, with much smaller numbers from other countries throughout the world. In the results sections, I have also added a brief analysis of the connection between country and open data mandates. There did not appear to be a significant correlation between place of publication and open data policy. I did not collect data on the scope of the journals for this study but I have included this facet as a potential area of further research.

3) How did the author define "LIS"? As appeared and/or classified in SCOPUS? This seems to be important to claim any finding in the context of LIS, and even within LIS, depending the nature and the focus of journals, each may have a different attitude toward open data policy. While what the author has presented in the results are meaningful, I think adding more contextual information about the journals would be another unique contribution to the field.

I used the categorization of Journal Citation Reports and SCOPUS to determine the journal list. I agree with the reviewer that these categories encompass a range of fields and research practices. I have added a more detailed description of the way that LIS is defined for the purposes of this research to the methods section, acknowledging that there are numerous sub-fields within the larger discipline. I have also included discussion toward the end of the paper about the ways in which this broad conception of LIS limits the findings and suggest a more narrow examination as an avenue of further research.

4) Lastly, it would be nice if the author can extend the discussion on the implications. Using all the findings, where should we go? What would the author suggest to do to make the existing situation better? Practical recommendations would be very helpful.

I have expanded the discussion to be more explicit about the impact of open data policies on data availability and provided recommendations for ways in which publishers using different models might move forward. I have also added a discussion of the limitations of the paper and suggestions for narrower research that might target more precise elements of open data policy that would benefit the discipline.

Reviewer 2 Report

The aim of this paper is to examine the information provided on public-facing websites of LIS journals in order to describe the extent and nature of open data guidance provided to prospective authors. Overall, the concept of this paper is promising although it suffers from several limitations (especially content-wise).

The main strengths of this paper are the following:

  • The title accurately reflects the content of this study.
  • The tables res are presented clearly and analyzed appropriately.
  • References used in this paper are up-to-date.

The abstract of the paper is complete and stand-alone. The author mentioned the objective as well as the practical implication of this research. Furthermore, the author highlighted the practical contribution of their paper.

The Introduction is focused. The author used the traditional structure in order to justify the research gap and the motivation as well as the value of this paper. The author presented the motivation of the paper and discussed about the theoretical and practical contribution of this paper.

The paper does not demonstrate an adequate understanding of the relevant literature in the field. The author should analyze the findings and research gaps from previous researchers. The author can present the main principles of open data based on the following papers:

  • Jaakkola, H., Mäkinen, T., & Eteläaho, A. (2014). Open data: opportunities and challenges. In Proceedings of the 15th International Conference on Computer Systems and Technologies (pp. 25-39).
  • Kitsios, F., & Kamariotou, M. (2019). Open Data Value Network and Business Models: Opportunities and Challenges. In 2019 IEEE 21st Conference on Business Informatics (CBI) (Vol. 1, pp. 296-302).
  • Kitsios, F., & Kamariotou, M. (2019). Service Dominant Logic and Digital Innovation from Open Data: Exploring Challenges and Opportunities. In International Conference on Innovation and Entrepreneurship (pp. 522-529). Academic Conferences International Limited.

The research on which the paper is based is well designed and the methods that have been employed are appropriate. The author presented clearly the results of the analysis. The findings are a good basis for discussion. However the author should clarify limitations and suggestions for future research.

Author Response

Response to Reviewer 2 Comments

I would like to thank the reviewer for their time and insightful review. My responses are below:

Point 1: The paper does not demonstrate an adequate understanding of the relevant literature in the field. The author should analyze the findings and research gaps from previous researchers. The author can present the main principles of open data based on the following papers.

  • Jaakkola, H., Mäkinen, T., & Eteläaho, A. (2014). Open data: opportunities and challenges. In Proceedings of the 15th International Conference on Computer Systems and Technologies (pp. 25-39).
  • Kitsios, F., & Kamariotou, M. (2019). Open Data Value Network and Business Models: Opportunities and Challenges. In 2019 IEEE 21st Conference on Business Informatics (CBI) (Vol. 1, pp. 296-302).
  • Kitsios, F., & Kamariotou, M. (2019). Service Dominant Logic and Digital Innovation from Open Data: Exploring Challenges and Opportunities. In International Conference on Innovation and Entrepreneurship (pp. 522-529). Academic Conferences International Limited.

In the paper, I have attempted to focus discussion of the literature on open data policies specific to academic publishing, journals in particular. I’ve provided a review of literature that addresses open data archiving as the practice relates to academic research, as well as the impact of journal policies on those practices. I feel that I have summarized the findings of that specific body of literature, and acknowledge the gaps specific to the discipline under study. There is certainly a plethora of research on open data in a wide variety of contexts. In order to set this study in the context of academic journal policies, though, I believe that the discussion of the literature would best focus on the nature and impact within the publishing and academic research environment.

Point 2: However the author should clarify limitations and suggestions for future research.

I have added a discussion of the limitations of the paper, including limitations in the reliability in the findings based on the method, the broad definition of LIS used in the paper, and the absence of discussion about compliance with open data policies. Some of these I have also suggested as areas of further research.

Reviewer 3 Report

An interesting and original paper, on a relevant topic. However, I would suggest to improve the description of the methodology and the discussion of the methodological approach. The issue is about assessment errors. In fact, we are doing the same kind of research (assessment of editorial policy) in a team where at least two members evaluate the same journal; and from time to time, the result is not the same, and we have to re-evaluate the journal. So, how did the author deal with this issue? How did he control the reliability of the assessment? Can he say something about the "uncertain part" of the evaluation?

No other comment on this useful paper.

Author Response

Response to Reviewer 3 Comments

I would like to thank the reviewer for their time and insightful review. My responses are below:

Point 1: I would suggest to improve the description of the methodology and the discussion of the methodological approach. The issue is about assessment errors. In fact, we are doing the same kind of research (assessment of editorial policy) in a team where at least two members evaluate the same journal; and from time to time, the result is not the same, and we have to re-evaluate the journal. So, how did the author deal with this issue? How did he control the reliability of the assessment? Can he say something about the "uncertain part" of the evaluation?

I agree completely that this section was rather sparsely described. I have added a more detailed description of the methods used. In this method, only one researcher examined the data. In order to establish some degree of intracoder reliability, the coding process was conducted twice, independently, after an initial review of the data. The codes were then compared against each other and again against the text extracted from journal websites. I acknowledge that this method does not carry the same degree of reliability as it would had the texts been coded by multiple team members and have flagged this in the paper as a limitation of the study.

Reviewer 4 Report

This comment is written for the article titled Open Data Policies Among Library and Information Science Journals. The article will require a major revision, focus on the Methodology and Results from sections, to meet publication requirements.

The article’s strengths and weaknesses can be analyzed from the scale of the data source, the analytics method, and the results and findings.

Data source (strong): It includes policies or guidance from websites of 201 LIS journals in June 2020.

Method (major improvement): The author states, “Multiple close readings of the textual information resulted in the identification of 12 codes” (P4, L191). Both “multiple” and “close” are vague and not a good fit for describing the research method. The author needs to describe, in a fair level of detail, approaches like how they read the policies (copy them locally or read them online), allocate reading time for each policy, and how to select and rank codes. The method section is the central part of content analysis, and it needs to be enhanced for this article.

Results and findings (improvement): The research splits LIS journals into OA and commercial ones, why? Is there a hypothesis behind the practice, and do the results approve/reject it? The recommendations in Table 2 are nice but the article seems to state that good policy will lead to good open data practice. Is there data or studies to support that statement?

Author Response

Response to Reviewer 4 Comments

I would like to thank the reviewer for their time and insightful review. My responses are below:

Point 1: Method (major improvement): The author states, “Multiple close readings of the textual information resulted in the identification of 12 codes” (P4, L191). Both “multiple” and “close” are vague and not a good fit for describing the research method. The author needs to describe, in a fair level of detail, approaches like how they read the policies (copy them locally or read them online), allocate reading time for each policy, and how to select and rank codes. The method section is the central part of content analysis, and it needs to be enhanced for this article.

I agree completely that this section was rather sparsely described. I have added a more detailed description of the methods used, including the coding process and clarification about both the semantic approach used to develop codes and the interpretive nature of code and theme development. 

Point 2: The research splits LIS journals into OA and commercial ones, why? Is there a hypothesis behind the practice, and do the results approve/reject it?

There is little research that I could find that links OA journals with open data policies (perhaps surprisingly), but there is some evidence that authors who have published in OA journals are more likely to make their data publicly available. I think there is also an intuitive connection between OA publishing and open data, but that is certainly an assumption. 

I think the characteristics of journals that have adopted open data policies sheds some light on how the practice has been adopted in the discipline. I have expanded sections of the paper to include discussion of the characteristics of journals within this sample, how LIS is defined for this article, and why information about the publishing models used in the sample might provide insight into good practices for open data policies. There isn’t a hypothesis underpinning the examination, but I think the publishing models used by these journals is part of the description of the nature of open data policies in the discipline, and I hope I have clarified that in my revisions.

Point 3: The recommendations in Table 2 are nice but the article seems to state that good policy will lead to good open data practice. Is there data or studies to support that statement?

I have added explicit reference to studies that do show strong connections between highly developed open data policies and public data archiving. This comment highlighted the need to distinguish between quality and quantity of data archiving practices, so I have tried to clarify my language. Data policies have been shown to lead to more data archiving, but there’s little information about the quality of those datasets. I’ve also added a brief discussion of this as an area of further research.

Round 2

Reviewer 4 Report

The revised version shows significant enhancements, I also appreciate the author's response that saves time spend on the second review. I have no more questions on the article for publication. 

Author Response

Thank you again for your time and thoughtful review.